# Phenotypic Switching of Naïve T Cells to Immune-Suppressive Treg-Like Cells by Mutant KRAS

**DOI:** 10.3390/jcm8101726

**Published:** 2019-10-18

**Authors:** Arjun Kalvala, Pierre Wallet, Lu Yang, Chongkai Wang, Haiqing Li, Arin Nam, Anusha Nathan, Isa Mambetsariev, Valeriy Poroyko, Hanlin Gao, Peiguo Chu, Martin Sattler, Andrea Bild, Edwin R. Manuel, Peter P. Lee, Mohit Kumar Jolly, Prakash Kulkarni, Ravi Salgia

**Affiliations:** 1Department of Medical Oncology and Therapeutics Research, City of Hope National Medical Center, Duarte, CA 91010, USA; akalvala@coh.org (A.K.); pwallet@coh.org (P.W.); chowang@coh.org (C.W.); anam@coh.org (A.N.); anathan@coh.org (A.N.); Imambetsariev@coh.org (I.M.); vporoyko@coh.org (V.P.); abild@coh.org (A.B.); pkulkarni@coh.org (P.K.); 2Department of Systems Biology, Beckman Research Institute, City of Hope, Duarte, CA 91010, USA; luyang@coh.org; 3Center for Informatics, City of Hope National Medical Center, Duarte, CA 91010, USA; hali@coh.org; 4Department of Computational and Quantitative Medicine, Beckman Research Institute, City of Hope National Medical Center, Duarte, CA 91010, USA; 5Fulgent Genetics, 4978 Santa Anita Avenue, Temple City, CA 91780, USA; harrygao@fulgentgenetics.com; 6Department of Pathology, City of Hope National Medical Center, Duarte, CA 91010, USA; PChu@coh.org; 7Department of Medical Oncology, Dana-Farber Cancer Institute, Boston, MA 02215, USA; Department of Medicine, Harvard Medical School, Boston, MA 02115, USA; Martin_Sattler@dfci.harvard.edu; 8Department of Hematology and Hematopoietic Cell Transplantation, City of Hope National Medical Center, Duarte, CA 91010, USA; EManuel@coh.org; 9Department of Immuno-Oncology, City of Hope National Medical Center, Duarte, CA 91010, USA; plee@coh.org; 10Center for BioSystems Science and Engineering, Indian Institute of Science, 560012 Bangalore, India; mkjolly.15@gmail.com

**Keywords:** KRAS, cell-extrinsic, Tregs, lung cancer, FOXP3, phenotypic switching

## Abstract

Oncogenic (mutant) Ras protein Kirsten rat sarcoma viral oncogene homolog (KRAS) promotes uncontrolled proliferation, altered metabolism, and loss of genome integrity in a cell-intrinsic manner. Here, we demonstrate that CD4^+^ T cells when incubated with tumor-derived exosomes from mutant (MT) KRAS non-small-cell lung cancer (NSCLC) cells, patient sera, or a mouse xenograft model, induce phenotypic conversion to FOXP3^+^ Treg-like cells that are immune-suppressive. Furthermore, transfecting T cells with MT KRAS cDNA alone induced phenotypic switching and mathematical modeling supported this conclusion. Single-cell sequencing identified the interferon pathway as the mechanism underlying the phenotypic switch. These observations highlight a novel cytokine-independent, cell-extrinsic role for KRAS in T cell phenotypic switching. Thus, targeting this new class of Tregs represents a unique therapeutic approach for NSCLC. Since KRAS is the most frequently mutated oncogene in a wide variety of cancers, the findings of this investigation are likely to be of broad interest and have a large scientific impact.

## 1. Introduction

The immune system is comprised of many biological structures and processes that protect the host against disease via adaptive and innate mechanisms. Adaptive immunity is mediated by T cell receptors, such as CD4, CD8, and CD25 that recognize specific antigens and neutralize them. Innate immunity, on the other hand, involves an immune response against antigens that is mediated by dendritic cells, macrophages, mast cells, neutrophils, basophils, and eosinophils that produce the cytokines that activate the adaptive immune system [1].

Regulatory T cells (Tregs) are immune-suppressive T cells that maintain tolerance to self-antigens and prevent autoimmune diseases. Tregs are characterized by the presence of the transcriptional regulator Forkhead box protein P3 (FOXP3) that plays an important role in immune suppression of effector T cell proliferation [2]. T cells follow one of two developmental pathways to enter the FOXP3^+^ Treg lineage. First, a developing thymocyte may recognize self-antigen presented within the thymus during T cell maturation and become a natural Treg (‘nTreg’). Alternatively, a conventional naïve CD4^+^ T cell may encounter a tumor-associated (‘self’) or tumor-specific (‘neo’) antigen in the tumor environment, become activated, and under the influence of an immunosuppressive tumor microenvironment, differentiate into an induced FOXP3+ (‘iTreg’) [3]. Within the tumor environment, however, Tregs also respond to context-dependent inflammatory signals (e.g., Th1, Th2, or Th17 inflammation). From these tumor environmental cues, Tregs are also capable of other functions, such as promotion of angiogenesis or metastasis, regulation of inflammation, and suppression of anti-tumor adaptive immune responses [4]. Since Tregs are frequently enriched within the tumor microenvironment, many emerging cancer therapeutic strategies involve depletion or modulation of Tregs, with the aim of enhancing anti-tumor immune responses [5,6,7,8].

Exosomes are small extracellular microvesicles (30–100 nm) of endocytic membrane origin that carry a repertoire of functional biomolecules, including genomic DNA, RNA, microRNA, and protein [9]. Upon secretion, exosomes, especially those derived from tumors (tumor-derived exosomes or TDEs), target other cells in the tumor microenvironment and deliver their payload, which can affect the targeted cell in multiple ways [10,11]. For example, H-RAS-mediated transformation of intestinal epithelial cells results in the emission of exosomes containing genomic DNA, HRAS oncoprotein, and transcript that can mediate transformation of normal cells [12]. TDEs are key players in epithelial to mesenchymal transition (EMT) and are notorious for orchestrating the invasion-metastasis cascade [9]. Additionally, TDEs from breast tumor cells that are resistant to antiestrogen drugs and metformin can mediate the transfer of the resistance phenotype to cells that are initially sensitive to these drugs [13]. In one of the most striking examples involving TDEs, transfer of mitochondrial DNA was shown to act as an oncogenic signal, promoting an exit from dormancy of therapy-induced cancer stem-like cells and leading to endocrine therapy resistance in OXPHOS-dependent breast cancer [14].

Taken together, these observations suggest that TDEs can mediate specific cell-to-cell communication to horizontally transfer information. By activating signaling pathways in the cells they fuse or interact with, TDEs can effect significant changes in recipient cells. Indeed, the concept of horizontal information transfer, previously thought to be confined to the prokaryotic world, is now gaining acceptance in biology, in general, and cancer biology, in particular [10,15].

In addition to their role in horizontal information transfer between tumor cells, emerging evidence has also uncovered a similar role of exosomes concerning immune regulation through direct interaction or by activating specific receptors present on immune regulatory cells. It is, therefore, not surprising that TDEs have been the focus of intense investigation, especially in cancer [16]. For example, when murine bone marrow precursor cells were incubated with TDEs, they caused expression of interleukin-6 (IL-6), which inhibited the differentiation of dendritic cells. This led to suppression of maturation, thereby causing a decrease in immune response [17]. Similarly, TDEs can activate macrophage-induced tumor invasion and metastasis function by inhibiting interferon gamma (IFNγ) and IL-16 and increasing secretion of IL-8 and C-C motif ligand 2 (CCL2) in target cells [18]. Finally, TDEs isolated from KRAS mutant (MT) human colon cancer cells can induce conventional CD4^+^ CD25^−^ T cells to convert to Tregs (CD4^+^ CD25^+^ FOXP3^+^) in a cell-extrinsic manner [19]. However, the phenomenon was found to be dependent on the secretion of IL-10 and TGF-β1 via the activation of the MEK-ERK-AP1 pathway. Indeed, disrupting KRAS signaling by knocking down MT KRAS using siRNA led to ~50% attenuation in the expression of the mRNAs expressing IL-10 and TGF-β1. Conversely, overexpression of MT KRAS in tumor cells harboring wild type (WT) KRAS resulted in an increase of mRNAs for the two cytokines [19].

In this manuscript, we interrogated the role of TDEs from non-small-cell lung cancer (NSCLC) in contributing to immune-suppression by horizontally transferring information to immune regulatory cells and inducing them to switch their phenotype. We show that TDEs isolated from MT KRAS NSCLC cells can convert naïve T cells to FOXP3^+^ Treg-like cells. Furthermore, we show that this phenomenon is independent of cytokine signaling, given that transfection of MT KRAS cDNA alone is sufficient to actuate the switch of naïve CD4^+^ T cells to a FOXP3^+^ phenotype with immune suppressive function, even in the absence of other biomolecules, such as cytokines. We present a mathematical model supporting these conclusions. Although the converted T cells express FOXP3 and are functional, they appear to be distinct from bona fide Tregs at the molecular level. Finally, we identified the interferon (IFN) pathway as the mechanism underlying the immune-protective to immune-suppressive switch.

## 2. Experimental Section

### 2.1. Study Design

The experiments described in this study used cell lines and live cells isolated from serum samples. For reproducibility and statistical analyses, experiments were each performed at least three times and in triplicate. For the in vivo studies, 4–5 mice were used per group. The details of each experiment are described below. Cut-off dates for data collection, data inclusion/exclusion criteria, outliers, randomization, blinding, etc., were not applicable to this study. All subjects gave their informed consent for inclusion before they participated in the study. The study was conducted in accordance with the Declaration of Helsinki, and the protocol was approved by the Ethics Committee of the Institutional Review Board (IRB) at City of Hope under IRB 17281.

### 2.2. Isolation of Tumor-Derived Exosomes (TDEs) from Patient Serum Samples

Serum from lung cancer patients with MT KRAS or WT KRAS were collected at the City of Hope National Medical Center under Institutional Review Board approval. TDEs were isolated using the Total Exosome Isolation Reagent (Invitrogen^™^ Waltham, MA, USA) following the manufacturer’s protocol.

### 2.3. Isolation and Transmission Electron Microscopy (TEM) of TDEs from Lung Cancer Cell Lines

TDEs were isolated from MT KRAS or WT KRAS cells as follows. Because cell culture media contains bovine pituitary extract enriched in microvesicles, all the experiments in which exosomes were collected or counted were performed using culture medium without fetal bovine serum (FBS). The culture medium was first centrifuged at 300× *g* for 10 min to remove cells and debris and subsequently centrifuged at 10,000× *g* for 45 min to remove large particles. Finally, the medium was ultracentrifuged twice at 110,000× *g* at 4 °C for 2 h in a Beckman Coulter Optima L-100XP ultracentrifuge to pellet the TDEs. TDEs were then suspended in a small volume of PBS and the samples were stored at −80 °C until used.

### 2.4. Nanosight Analysis and Concentration Determination

Nanoparticle tracking analysis was used to determine TDE concentration. TDE samples were diluted 1:10 in PBS and visualized with the NanoSight NS300 nanoparticles detector (Malvern, Westborough, MA, USA). The preparations were introduced into the sample chamber of the instrument equipped with a 635 nm laser. All samples were diluted to give counts in the linear range of the instrument (up to 7 × 10^8^ per mL). The particles in the laser beam undergo Brownian motion and videos of these particle movements are recorded. The Nanosight Tracking Analysis (NTA) 2.3 software (Malvern Analytical, Malver, PA 19355, USA) then analyzes the video and determines the particle concentration and the size distribution of the particles. Three videos of 30 s duration were recorded for each sample at appropriate dilutions with a shutter speed setting of 1500 (exposure time 30 ms) and camera gain of 560. The detection threshold was set at 6 and at least 1000 tracks were analyzed for each video.

### 2.5. Genomic and TDE DNA Isolation

Total DNA from cells was isolated using the DNeasy Blood and Tissue Kit (Qiagen, Germantown, MD 20874, USA; Qiagen, hilden, Germany). TDE DNA was isolated from the serum-depleted cell culture supernatants treated with proteinase K, lysis buffer, and precipitated with ethanol (100%) followed by heat inactivation at 56 °C.

### 2.6. Isolation of CD4^+^ T and Naïve CD4^+^ CD25^−^ T Cells from Donor PBMCs

PBMCs from healthy donors were processed for isolation of CD4^+^, and naïve CD4^+^ T cells (CD4^+^ CD25^−^ T) cells using Histopaque (Sigma Aldrich, Munich, Germany). Briefly, 5 mL of donor blood was diluted with PBS and upon centrifugation over Histopaque solution, PBMCs were isolated. Approximately, 1 × 10^7^ mL of PBMCs were used for isolation of CD4^+^ T cells using the MojoSort™ Human CD4^+^ T Cell Isolation Kit (catalog; 480009). For isolation of naïve CD4^+^ CD25^−^ T cells, the MojoSort^TM^ Human CD4 naïve T cell isolation kit (catalog; 480041) (BioLegend, San Diego, CA, USA) was used. 

### 2.7. Isolation of Human CD4^+^ CD127^low^ CD25^+^ Regulatory T Cells from Donor PBMCs

PBMCs from healthy donors were processed for isolation of CD4^+^ CD127^low^ CD25^+^ Regulatory T cells using Histopaque (Sigma Aldrich, Munich, Germany). Briefly, 5 mL of donor blood was diluted with PBS and upon centrifugation over Histopaque solution, PBMCs were isolated. Approximately, 1 × 10^7^ mL of PBMCs were used for isolation of CD4^+^ CD127^low^ CD25^+^ Regulatory T cells using the EasySep™ CD4^+^ CD127^low^ CD25^+^ Human Regulatory T Cell Isolation Kit (STEMCELL Technologies, Cambridge, MA, USA) following the manufacturer’s protocol.

### 2.8. Cell Culture and Transfection

The human NSCLC cell lines A549, H358, H460, and H1299 were maintained in complete growth medium containing RPMI from (Life Technologies, Camarillo, CA, USA) with 10% FBS and antibiotics penicillin and streptomycin. The CRISPR/Cas9 plasmid encoding the target wild type sgKRAS sequence was purchased from Addgene. CRISPR/Cas9 plasmid at 2 µg concentration was transfected by Turbofectin 8.0 following the protocol from OriGene (Rockville, MD, USA).

### 2.9. Site-Directed Mutagenesis and TOPO^®^ TA Cloning

Plasmid pBabe-KRas WT KRAS (Plasmid# 75282) and pBabe-KRAS G12D (Plasmid # 58902) were purchased from Addgene. The pBabe-KRAS point mutation Q61H was created by using the QuikChange II Site-Directed Mutagenesis Kit (Agilent Technologies) following the recommended protocol. For TOPO^®^ TA Cloning, pCR™4-TOPO™ Vector kit was purchased from (Thermo Fisher Scientific, Waltham, MA, USA) and the PCR product was cloned into the TOPO vector following the manufacturer’s protocol. The pCMV-AC- KRAS GFP fusion plasmid was purchased from OriGene (Rockville, MD, USA) and used as a template for construction of Q61H KRAS mutation by the site-directed mutagenesis.

### 2.10. Western Blotting

The TDE pellet was dissolved in RIPA buffer and quantified using the BCA Protein Assay method (ThermoFisher Scientific, Waltham, MA, USA). Reducing buffer was added to the samples and the proteins were analyzed by SDS-Mini Protean TGX gels (Bio-Rad, Irvine, CA, USA) and transferred to Immobilon-P PVDF Membrane (Millipore Sigma, Darmstadt, Germany). The membrane was then incubated with target antibodies at 4 °C overnight. After washing the membrane with tris-buffered saline (TBS) and Tween20 (TBS-T) three times for 30 min, the membrane was incubated with secondary antibodies for 1 h at room temperature. After washing five times, 10 min each time, a chemiluminescent detection system (Bio-Rad Clarity western ECL substrate) was used to detect the secondary antibody. Finally, the membranes were exposed to X-ray films to detect the proteins of interest. Antibodies used were a mouse monoclonal KRAS antibody (Santa Cruz Biotechnology, Dallas,TX, USA) and for detection of exosomes, anti-CD9, anti-CD63, and anti-CD81 antibodies (System Biosciences, Palo Alto, CA, USA).

### 2.11. Incubation of Exosomes with CD4^+^ T and Naïve CD4^+^ CD25^−^ T Cells

CD4 T cells at a density of 2 × 10^6^ cells/well in 24-well plates were seeded in RPMI medium without FBS. Naïve CD4^+^ CD25^−^ T cells at a density of 1 × 10^6^ cells/well in 24-well plates were seeded in RPMI medium without FBS. After 3–4 h, TDEs isolated from control, WT KRAS TDEs and Q61H mutant KRAS lung cancer cells were incubated with CD4 and naïve CD4 T cells for 48 h and analyzed by flow cytometry.

### 2.12. Transfecting CD4^+^ and Naïve CD4^+^ CD25^−^ T Cells with KRAS cDNAs

CD4^+^ T cells at density of 3 × 10^6^ mL and naïve CD4^+^ CD25^−^ T cells at a density of 1.5 × 10^6^ mL were seeded in 24-well plates, and 3–4 h later were transfected with 2 µg of pBabe KRAS WT and pBabe KRAS G12D or pBabe KRAS Q61H plasmid DNA by electroporation using the Amaxa^®^ Human T Cell Nucleofector^®^ Kit (Lonza, Basel, Switzerland) following the manufacturer’s protocol. Twenty-four hours post transfection, 1 µg/mL of puromycin was added, and after 48 h, the cells were analyzed by flow cytometry.

### 2.13. Cell Surface and Nuclear Staining

The cells were stained with anti-human monoclonal antibody (mAb) CD4–FITC clone RPA-T4 (BioLegend, San Diego, CA, USA). For nuclear staining, the cells were fixed, and permeabilized using the True Nuclear Transcription Factor Buffer (BioLegend, San Diego, CA, USA). The anti-human mAb FOXP3-PE antibody (BioLegend, San Diego, CA, USA) was used for staining intranuclear FOXP3. Flow cytometry data were analyzed using FlowJo.

### 2.14. TDE Flow Cytometry Analysis

TDEs isolated from the NSCLC cells and patient serum samples were directly analyzed for intracellular cytokine IL-10 and TGF-β1 expression by Attune NxT flow cytometer. For this purpose, TDEs in 100 μL final volume of PBS were stained using the PE anti-human CD9 antibody (Clone HI9a) (BioLegend, San Diego, CA, USA) and incubated for 1 h at 4 °C. The TDEs were then washed and stained for intracellular cytokines using the PE/Cy7 anti-human IL-10 antibody (clone JES3-9D7) and Brilliant Violet 421™ anti-human LAP (TGF-β1) antibody (TW4-2F8) (BioLegend, San Diego, CA, USA). TDEs isolated from mice serum were stained with APC anti-human CD63 Antibody (Clone H5C6) (BioLegend, San Diego, CA, USA) and FITC anti-human KRAS antibody (Catalog#: OACA01930, Aviva Systems Bio, CA, USA).

### 2.15. Immune Treg Phenotype Cell Metabolism Assay

To assay for the Treg phenotype cell metabolism assay, CD4^+^ CD127^low^ CD25^+^ regulatory T cells or naïve CD4^+^ CD25^−^ T cells alone or after incubation with MT KRAS TDEs were seeded at cell density of 20,000 cells in 50 µL/well in a PM-M1 Microplate™ (Biolog, Hayward, CA, USA) for analysis of carbon metabolism. Briefly, the regulatory T cells, naïve CD4^+^ CD25^−^ T cells alone or after incubation with MT KRAS TDEs for 24–48 h were mixed with 10 µL of Dye MB and the plates were incubated in the OmniLog^®^ incubator. Cell metabolism of the immune Tregs was calculated based on the redox dye intensity in units. Mutant and WT KRAS NSCLC were also assessed for carbon metabolism using the PM-M1 Microplate™ (Biolog, Hayward, CA, USA).

### 2.16. IncuCyte^®^ S3 Cell Count Proliferation Assay

Immune cell proliferation assay was performed using the IncuCyte^®^ Cell Count Live Cell Imaging System S3 (IncuCyte, Ann Arbor, MI, USA). Briefly, naïve CD4^+^ CD25^−^ T cells at density of 1 × 10^4^ cells/well were incubated with MT KRAS or WT KRAS TDEs in 96-well plates for 24 h and cell proliferation was analyzed by counting the cells in real time. For CD4^+^ CD127^low^ CD25^+^ Treg proliferation assay, 1.5 × 10^4^ were incubated with MT KRAS TDEs or WT KRAS TDEs in a 96-well plate for 24 h, and cell proliferation was analyzed by counting the cells using in real time IncuCyte ^®^ S3 Live Cell Imaging System. 

### 2.17. In vitro Immune Suppression Assay

Naïve CD4^+^ CD25^−^ T cells at density of 1 × 10^6^ /well were seeded in a 24-well plate with MT KRAS or WT KRAS TDEs for 24 h. CD4^+^ CD127^low^ CD25^+^ Tregs at a density of 3–5 × 10^5^ well were seeded in a 24-well plate. In a separate well, naïve CD4^+^ CD25^−^ T cells at density of 1 × 10^6^ well were pre-stained with CellTrace™ CFSE Proliferation Kit (ThermoFisher Scientific, MA). After 24 h, the naïve CD4 T carboxyfluorescein succinimidyl ester (CFSE) pre-stained cells were then added to the naïve CD4^+^ CD25^−^ T cells that were pre-incubated with WT or MT KRAS TDEs. After 24 h, cells were fixed and permeabilized and analyzed by flow cytometry. For flow cytometry analysis, cells were analyzed by fixing and staining with intranuclear antihuman mAb FOXP3-PE antibody (BioLegend, San Diego, CA, USA) and counted for cells that are positive for CFSE dye intensity. CD4^+^ CD127^low^ CD25^+^ Tregs or naïve CD4^+^ CD25^−^ T cells alone were used as control. 

### 2.18. Salmonella typhimurium Host Strains

YS1646 was purchased from ATCC^®^ (202165, Manassas, VA, USA). The bioluminescence expression vector for bacterial plasmid pAKlux2 was purchased from Addgene (Watertown, MA, USA) [20]. Bacteria were grown to the OD = 0.7, washed three times with equal volumes of sterile PBS and diluted in PBS to the final concentration of 1 × 10^5^ CFU/μL. To inactivate bacteria, the aliquoted preparation was incubated at 70 °C for 1 h.

### 2.19. NSCLC Xenograft Mouse Model

Human NSCLC cells A549 (5 × 10^6^) and H358 (5 × 10^6^ cells) were injected, in left and right flanks of NSG mice (*n* = 12), respectively, to create a two-sided xenograft model. After injection, tumor size was monitored three times/week and tumor volume ((length × width)^2^ × 0.5) was calculated. When tumor volumes reached 100 mm^3^ mice were randomly assigned in 3 groups (*n* = 4) to receive three consecutive daily retro-orbital injections (50 μL) of sterile PBS (Group 1), or 5 × 10^6^ CFU preparation of heat-inactivated (Group 2) and live bioluminescent *Salmonella typhimurium* YS1646 (Group 3). After injection, the tumor growth was monitored for 21 days. Animals were then euthanized and blood was collected. Mice PBMC fractions were isolated using Histopaque (Sigma Aldrich, Munich, Germany) by collecting the PBMC supernatant after centrifugation at 2000 rpm for 20 min. Exosomes from mice serum were isolated using the total Exosome Isolation Reagent (Invitrogen^™^, Waltham, MA, USA) following the manufacturer’s protocol.

### 2.20. Ex vivo Human MT KRAS Exosomes and Immune Phenotype Conversion Assay

NSCLC xenograft serum exosomes containing MT KRAS from the control PBS (Group 1), or heat-inactivated *Salmonella typhimurium* YS1646 (Group 2)- or live bioluminescent *Salmonella typhimurium* YS1646 (Group 3)-treated mice were incubated with the naïve CD4^+^ CD25^−^ T cells isolated from healthy donor PBMCs at a density of 1 × 10^6^ well in a 24-well plate for 48 h. The cells were stained with antihuman mAb CD4–FITC clone RPA-T4 (BioLegend, San Diego, CA, USA). For nuclear staining, the cells were fixed, permeabilized using the True Nuclear Transcription Factor Buffer (BioLegend, San Diego, CA, USA). The anti-human mAb FOXP3-PE antibody (BioLegend, San Diego, CA, USA) was used.

### 2.21. Infection with Salmonella typhimurium

Naïve CD4^+^ CD25^−^ T cells at a density of 5 × 10^5^ well were incubated with WT KRAS or MT KRAS TDEs for 24 h. The live “T37” or heat-inactivated “T70” bioluminescent *Salmonella typhimurium* preparations (1 × 10^5^ CFU/µL) were diluted in PBS (1:10) and co-cultured with the naïve CD4 T cells that were pre-incubated with TDEs and after 24 h, the cells were fixed and stained for CD4^+^ and FOXP3^+^ and analyzed by flow cytometry.

### 2.22. In vitro Salmonella typhimurium Human MT KRAS TDEs and Immune Phenotype Conversion Assay

Naïve CD4^+^ CD25^−^ T cells isolated from healthy donor PBMCs were seeded in 24-well plate at a density of 5 × 10^5^ cells/well. WT KRAS or MT KRAS TDEs were incubated with the naïve CD4^+^ CD25^−^ T cells. Next, we diluted the preparation (1 × 10^5^ CFU/µL) of heat-inactivated “T70” or live “T37” bioluminescent *Salmonella typhimurium* in 1:10 PBS and then incubated with the naïve CD4^+^ CD25^−^ T cells pre-incubated with the WT KRAS or MT KRAS TDEs. PBS was used as control in the experiments. After 24 h, the cells were fixed and stained by anti-human mAb FOXP3-PE antibody (BioLegend, CA, USA) and anti-human mAb CD4–FITC clone RPA-T4 (BioLegend, San Diego, CA, USA). The flow cytometry data analysis was carried out using FlowJo^®^ (v10.6.1, Ashland, OR, USA).

### 2.23. Single-Cell Analysis

Sequences from 10× genomics platform were de-multiplexed and aligned to genome build hg19 using CellRanger 3.0.1. (https://support.10xgenomics.com/single-cell-gene-expression/software/pipelines/latest/what-is-cell-ranger). To balance the size differences of the Treg group to KRAS MT and WT groups, 1000 cells were randomly selected from MT and WT groups for the clustering analysis over all three groups. Data were then imported into Seurat package version 2.3.4 for further filtering, variable gene selection, dimensional reduction and clustering analysis. We filtered out cells that have unique gene counts less than 200 or over 2000 and high mitochondrial expression (>10% mitochondria counts). After normalization, 1619 genes were identified as variable genes for PCA. Using embedded jackstraw analysis in the Seurat package, we identify the first 10 statistically significant principal components from a previous PCA step to be used in the non-linear dimensional reduction clustering analysis (tSNE). Marker genes for each group were identified by performing differential expression analysis in Seurat, with the exclusion of genes that had less than 25% detection percentage in either of the two comparison groups.

### 2.24. Fractal Dimension (FD) and Lacunarity (LC) Analysis

Immunohistochemistry (IHC) staining images with three markers (DAPI, CD4 and FOXP3) were scanned for the presence of both tumor cells and surrounding stroma and immune cells in the tissue microenvironment. The DAPI-stained nuclei were used to identify the overall cells in the tissue. CD4 and FOXP3^+^ staining images were used to identify T-helper cell and Treg cell distribution. IHC staining images were segmented to identify the cells on the tissue slides using Fiji/ImageJ ver 1.52 [21]. In this study, we measured the Fractal Dimension (FD) and Lacunarity (LC) of the merged binary IHC images using box counting scan method implemented in the FracLac plugin ver2016Apr120248a502 [22] from Fiji/ImageJ version 1.52 [21]. A total of 558 IHC staining images with 95% tissue coverage were collected from 15 lung cancer patients. A total of 264 slides were from KRAS mutant tumor samples, and 294 slides were from wild type tumor samples. The difference of FD and lacunarity between the KRAS mutant and wild type samples were tested using Mann–Whitney *U*-test. The *p*-value indicates the level of distance between the two data sets. All tests were performed using R (version 3.5.1, R Foundation, Vienna, Austria) [22]. The test results show that the mean of the KRAS mutant FD is significantly higher than the mean of wild type KRAS FD (*p*-value = 7.152 × 10^5^). The mean of the KRAS mutant lacunarity is significantly less than the mean of the wild type KRAS lacunarity (*p*-value = 3.297 × 10^6^).

### 2.25. Statistical Analyses

A two-tailed Wilcoxon rank-sum test was used to test the null hypothesis that two sets of measurements were drawn from the same distribution. This was used instead of the paired Student’s *t*-test as the populations could not be assumed to be normally distributed.

## 3. Results

### 3.1. Isolation and Characterization of TDEs from NSCLC Cell Lines

TDEs from supernatants of MT KRAS and WT KRAS NSCLC cell cultures were isolated as described in the Materials and Methods. To characterize the TDE preparations, we subjected them to Transmission Electron Microscopy (TEM) and estimated their size and concentration using the Nanosight NS300 system. This technique uses the properties of both light scattering and Brownian motion to calculate particle size and concentration. A representative TEM (Appendix A) showed TDEs had uniform appearance and a mean size of 100 nm (Appendix A). To determine the biochemical nature of the exosomal cargo, we isolated high-molecular-weight genomic DNA from A549, H460, H358 and H1993 human NSCLCs, as well as from TDEs isolated from the respective cell lines. Using KRAS-specific primers, the DNA samples were PCR-amplified and resolved by agarose gel electrophoresis. Both cellular DNA and exosomal DNA yielded amplicons of the expected size corresponding to the KRAS sequence (Appendix A). To confirm the identity of the PCR-amplified DNA fragments, we subjected them to Sanger sequencing, which revealed the presence of the Q61H mutation and a 100% match to the KRAS sequence (Appendix A). We also identified the KRAS mutations G12S, G12C, and Q61H in both cellular genomic DNA and exosomal DNA isolated from these cells (Appendix A). Having confirmed the presence of the mutant KRAS DNA, next we asked if the KRAS protein was present in the TDEs. TDE preparations were lysed with lysis buffer, and total protein was isolated and subjected to Western blotting, then probed with a KRAS-specific antibody. KRAS protein was indeed detected in isolated TDEs (Appendix A).

### 3.2. MT KRAS Converts Naïve T Cells to CD4^+^ FOXP3^+^ Treg-Like Cells 

Total CD4^+^ T cells isolated from healthy donor peripheral blood mononuclear cells (PBMCs) were incubated for 48 h with WT KRAS or Q61H MT KRAS TDEs isolated from H460 cell lines. The frequency of FOXP3^+^ cells, a hallmark of Tregs, was analyzed by flow cytometry. We found that, compared to untreated CD4^+^ T cells, co-culturing CD4^+^ T cells with WT TDEs resulted in a ~1.5-fold increase in conversion to CD4^+^ FOXP3^+^ (Figure 1A). However, when incubated with Q61H MT KRAS TDEs, we observed a ~5.8-fold increase in the conversion to CD4^+^ FOXP3^+^ cells (*p* = 8.2 × 10^−3^), suggesting that MT KRAS TDEs efficiently mediate phenotypic switching of T cells to Treg-like cells. We also isolated naïve CD4^+^ CD25^−^ FOXP3^−^ T cells and incubated them with either WT KRAS or Q61H MT KRAS TDEs. Compared to untreated naïve CD4^+^ T cells, incubating naïve CD4^+^ T cells with WT TDEs, which resulted in a 1.8-fold conversion to CD4^+^ FOXP3^+^ (Figure 1A). However, when incubated with Q61H MT KRAS TDEs, we observed a ~6-fold increase (*p* = 3.9 × 10^−3^) in the conversion to CD4^+^ FOXP3^+^ cells.

### 3.3. MT KRAS TDEs from NSCLC Patient Serum Can Efficiently Switch the T Cell Phenotype 

Having demonstrated that KRAS MT TDEs from NSCLC cell lines can efficiently switch the T cell phenotype from protective to suppressive, we next asked whether TDEs isolated from patient serum would also be able to convert the T cells. To this end, we isolated TDEs from NSCLC patients that harbored either a WT or MT version of the KRAS gene, incubated them with naïve CD4^+^ CD25^−^ T cells as described above and determined the frequency of CD4^+^ FOXP3^+^ cells by flow cytometry. Compared to WT KRAS, MT KRAS TDEs isolated from different patient sera resulted in a significant (3- to 10-fold) increase in the conversion of T cells to Treg-like cells (Figure 1B), similar to our finding with TDEs from cell lines. 

### 3.4. Expansion of the Treg-Like Population is Due to Phenotypic Switching Rather Than Increased Proliferation of Pre-Existing Tregs

Although the co-culturing experiments indicated phenotypic conversion, it may be argued that the increase in the Treg-like population could simply be due to increased proliferation of rare Treg-like cells that pre-existed in the CD4^+^ population, rather than a true phenotypic conversion of FOXP3^−^ T cells. To rule out this possibility, we constructed a mathematical model that considers three possibilities to account for the conversion (Figure 2A). Model I considers the role of mutation-driven changes in the Treg-like population (i.e., MT KRAS DNA being incorporated into the genome of the recipient cell). Model II considers enhanced proliferation of pre-existing cells. Model III considers phenotypic plasticity between Tregs and non-Tregs, such that the conversion into Treg-like cells is enhanced by MT KRAS DNA. In each model described below, X denotes CD4^+^ (or non-Tregs) and Y denotes Tregs. Details of the parameters and model formulations are provided in the Appendix A.

Among all the models, model III (plasticity between Tregs and non-Tregs) is most likely to reflect a saturating effect, in terms of number of non-Tregs that convert to a Treg-like phenotype upon exposure to MT KRAS TDEs, unlike models I and II. To falsify the predictions, we performed two sets of experiments. First, we isolated naïve CD4^+^ CD25^−^ FOXP3^−^ cells from healthy donor PBMCs, incubated them for 24 h with either WT KRAS or MT KRAS TDEs, and measured naïve CD4^+^ T cell proliferation by counting the cells. We detected no significant increase in cell number of naïve CD4^+^ T cell in the presence of MT KRAS compared to WT KRAS (Figure 2B). Second, naïve CD4^+^ T cells were incubated with either WT KRAS or MT KRAS and stained with a Ki-67 antibody. The stained cells were then analyzed by flow cytometry with gating for CD4^+^ FOXP3^+^ Tregs. Cells incubated with MT KRAS showed no significant increase in Ki-67 expression (Appendix A). Taken together, these results suggest that the increase in CD4^+^ FOXP3^+^ Treg-like cell population was indeed due to phenotypic conversion rather than an increase in proliferation of pre-existing Treg-like cells. Alternatively, these results may be interpreted as an unchanged proliferation rate in the Treg-like cells in comparison to control or CD4+ T cells incubated with WT KRAS TDEs rather than an evidence of no increased proliferation of pre-existing Treg-like cells.

### 3.5. MT KRAS MT TDEs from NSCLC Patient Serum Do Not Increase Proliferation FOXP3^+^ Tregs

Next, we isolated CD4^+^ CD25^+^ CD127^low^ (FOXP3+) Tregs from healthy donors and incubated them with WT and MT KRAS TDEs from NSCLC patient serum samples for 24 h. We observed no increase in the percentage of FOXP3^+^ population as measured by flow cytometry (Figure 2C). We further analyzed cell proliferation by co-culturing MT KRAS or WT KRAS exosomes with CD4^+^ FOXP3^+^ CD127^low^ (FOXP3^+^) Treg cells and counting the cells. Again, we did not observe a significant increase in cell proliferation, further supporting the hypothesis that expansion of the Treg-like population in the presence of MT KRAS TDEs is due to phenotypic conversion rather than proliferation of pre-existing Treg-like cells, as was also predicted by mathematical modeling (Appendix A).

### 3.6. MT KRAS cDNA, in the Absence of Other Biomolecules, Is Sufficient to Convert T Cells to Tregs

Because isolated exosomes carry many biomolecules in addition to DNA, it is possible that the observed phenotypic switching is at least partially attributable to these other molecules such as cytokines, as reported previously [19]. To rule out this possibility, we transfected naïve T cells by electroporation with a plasmid DNA construct in which the full-length KRAS cDNA harboring the G12D or Q61H was cloned downstream of a basic CMV promoter. Conversion to a Treg-like phenotype was measured 24 h later by flow cytometry, as described above. A robust 27-fold increase in the CD4^+^ FOXP3^+^ T cell population was observed with the MT KRAS as compared to CD4^+^ T cells transfected with the WT KRAS plasmid. These data indicate that overexpression of MT KRAS protein resulting from transcription/translation of the KRAS cDNA/mRNA is sufficient to actuate the phenotypic switch (Figure 3A and Appendix A). 

To interrogate the potential contribution of secretory immune-suppressive cytokines in phenotypic conversion of naïve CD4^+^CD25^−^ T cells, we determined the expression of IL-10 and TGF-β1 in WT and MT KRAS TDEs from NSCLC cell lines by flow cytometry. We found that both IL-10 and TGF-β1 were significantly downregulated in MT KRAS TDEs compared to WT KRAS TDEs, further supporting the argument that the observed phenotypic conversion is not dependent on these cytokines (Figure 3B,C and Appendix A). We also quantified IL-10 in MT KRAS TDEs by enzyme-linked immunosorbent assay, but IL-10 was not present at detectable levels in these samples (Figure 4A).

### 3.7. Converted T Cells Can Suppress T Cell Proliferation

To characterize the converted Treg-like cells functionally, we incubated naïve CD4^+^ CD25^−^ T cells isolated from donor PBMCs with MT KRAS or WT KRAS TDEs for 24 h and then added naïve carboxyfluorescein succinimidyl ester (CFSE)-labeled CD4^+^ CD25^−^ T cells. Bona fide Tregs isolated from the same donors and characterized by CD4^+^ CD25^hi^ CD127^low^ were used as a positive control. We found that cells incubated with MT KRAS TDEs showed suppression of CD4^+^ naïve T cell proliferation that was comparable to the authentic Tregs (Figure 4B). However, cells incubated with WT KRAS TDEs showed a significantly impaired ability to suppress the proliferation of naïve CD4^+^ CD25^−^ T cells. Moreover, CD4 naïve T cells co-cultured with MT KRAS TDEs secrete more IL-10 compared to CD4^+^ naïve T cells cultured with WT KRAS TDEs (Figure 4A). Considered together, these observations provide compelling evidence that the converted T cells are indeed Treg-like cells that are functionally active. 

### 3.8. Converted Treg-Like Cells Display a Metabolic Profile Similar to that of Bona Fide FOXP3^+^ Tregs 

To further validate the converted CD4^+^ FOXP3^+^ cells, we discerned the metabolic profile of TDE-induced Treg-like cells and compared this profile to bona fide FOXP3^+^ Tregs. The latter have been shown to function in low-glucose environments rich in lactate [23]. 

Naïve CD4^+^ CD25^−^ T cells were used as a negative control, and bona fide Tregs were used as a positive control. Naïve CD4^+^ CD25^−^ T cells were incubated with MT KRAS TDEs from NSCLC patient sera in a Biolog microplate consisting of a pre-arrayed set of 96 different carbon and energy sources in each well, and the plates were incubated in an OmniLog incubator at 37 °C. The rates of carbon utilization and energy production were measured using the proprietary redox dye. Based on the percent intensity of the redox dye coloration due to the generation of energy-rich NADH, we determined that naïve CD4^+^ CD25^−^ T cells had high levels of glucose-6-phosphate uptake. Utilization of glucose-6-phosphate and other carbon sources were significantly reduced in bona fide Tregs and naïve CD4^+^ CD25^−^ T cells incubated with MT KRAS TDEs in comparison to naïve CD4^+^ T cells alone (Appendix A), suggesting that the converted cells exhibit a Treg-like phenotype (Appendix A). 

### 3.9. TDEs Isolated from Mice Harboring Exponentially Growing Tumors Can Convert T Cells to Treg-Like Cells More Efficiently Compared to Those Derived from Attenuated Tumors

Previous studies have shown that, among other mechanisms, Salmonella activates CD8^+^ T cell immune responses to eliminate tumor cells [24,25]. In mouse spleens infected with Salmonella enterica serovar Typhimurium, analysis of Tregs characterized by CD4^+^ CD25^hi^ CD127^low^ revealed that the decrease in tumor growth depended on lipopolysaccharides and lipoproteins [26]. Therefore, we asked if attenuated Salmonella would show a similar inhibition in MT KRAS-actuated phenotypic switching of T cells. For this purpose, NSG mice were engrafted with A549 (G12D) on the left flank and H358 (G12C) MT KRAS NSCLC cells on the right flank, and then infected with live bioluminescent Salmonella bacteria [20]. Colonization was demonstrated by whole-body imaging, and the bacteria were found to successfully colonize both types of tumor (A549 and H358) at day seven post-infection. We observed that bacterial colonization significantly suppressed formation of A549 tumors (on the left) but did not significantly affect the H358 tumors (on the right). In contrast, in control mice that were injected with phosphate-buffered saline (PBS) or heat-inactivated bacteria, no suppression of tumor growth was observed (Figure 5A). 

At the end of the experiment, the mice were euthanized, sera were collected using Histopaque solution, and exosomes were isolated from the serum (Figure 5B) and incubated with naïve CD4^+^ CD25^−^ T cells isolated from donor PBMCs. MT KRAS exosomes showed a ~6-fold increase in the conversion to CD4^+^ FOXP3^+^ Treg-like cells compared to naïve CD4 T cells (Figure 5C). We further determined the effect of live or heat-inactivated bioluminescent Salmonella on naïve CD4^+^ CD25^−^ T cells co-cultured with WT or MT KRAS TDEs in vitro. Naïve CD4^+^ CD25^−^ T cells were incubated with WT KRAS or MT KRAS TDEs for 24 h and then infected with live or heat-inactivated bacteria for 24 h. The cells were then fixed and stained for CD4^+^ and FOXP3^+^ and analyzed by flow cytometry. Naïve CD4^+^ cells incubated with MT KRAS TDEs isolated from human lung cancer cells were used as a control. We observed a decrease in Treg-like cell conversion after infection with live bioluminescent Salmonella compared to MT KRAS TDEs (Figure 5D). Together, these experiments suggest that live Salmonella bacteria decrease the rate of tumor growth and thereby affect the number of TDEs in circulation. 

### 3.10. Single-Cell Sequencing Identified Interferon Signaling as the Driver in MT KRAS-Mediated Phenotypic Conversion to Treg-Like Cells

To validate the identity of the converted FOXP3^+^ cells, we performed single-cell sequencing (scRNA-Seq). The scRNA-Seq technique has several advantages over bulk RNA-Seq, including analysis of distinct cell types with subtle differences and exclusion of unrelated cells affecting bulk measurement [27]. Three groups of cells were profiled. Group 1 (control Group), comprised of naïve CD4^+^ T cells incubated with TDEs from WT KRAS NSCLC cells. Group 2 included naïve CD4^+^ T cells incubated with TDEs from MT KRAS NSCLC cells, and Group 3 included enriched Tregs isolated from donor PBMCs served as a positive control. To identify gene expression profiles related to conversion of naïve T cells induced by MT KRAS TDEs, we first used a t-SNE clustering approach to observe global transcriptional differences in non-Treg versus bona fide Tregs (Figure 6A). Tregs, CD4^+^ naïve T cells treated with WT KRAS TDEs, and CD4^+^ naïve T cells treated with MT KRAS TDEs are clearly clustered independently from each other, suggesting discrete populations of cells and phenotypes. Treg and CD4^+^ naïve cells treated with MT KRAS shared a proportion of differentially expressed genes, while naïve cells treated with WT KRAS did not. Further, CD4^+^ naïve cells treated with MT KRAS TDEs showed a strong IFN pathway signature with upregulation of several IFN-stimulated genes (Figure 6B). In particular, CD4^+^ naïve cells treated with MT KRAS TDEs and Tregs showed upregulation for IRF7 and EIF2AK2 (PKR), genes that are important in regulation of the IFN pathway (Appendix A). We calculated the expression level of each upregulated gene observed in the CD4^+^ naïve cells treated with MT KRAS TDEs using the average gene expression of Treg cells as a threshold (Treg cells without detectable gene expression were excluded). We found that 11.2% and 16.9% of the CD4^+^ naïve cells treated with MT KRAS TDEs had upregulated IRF7 and PKR gene expression, respectively, compared to Tregs control (Figure 6C). However, a range of 12.9% to 65.6% of those cells showed upregulated expression of ISGs compared to the expression pattern defined in the Tregs control. IRF7 induces type I IFN secretion, and IFN can induce CD4^+^ naïve T cells to Tregs [28,29,30]. The proportion of cells expressing IRF7 compared to Tregs is interestingly close to the proportion of converted Treg-like cells found by flow cytometry. Therefore, we hypothesize that CD4^+^ Treg-like cells converted by MT KRAS secrete type I IFN via the upregulation of IRF7, and other non-converted naïve CD4^+^ T cells respond to the secreted IFN. Altogether, these results demonstrate that MT KRAS convert naïve CD4^+^ T cells to Tregs-like with immune-suppressive functions. This conversion induces a distinct gene signature related to IFN signaling. 

### 3.11. NSCLC Tumor Microenvironment is Enriched in Tregs

Having demonstrated that MT KRAS contributes to immunosuppression via a cell-extrinsic mechanism, we next assayed the tumor microenvironment for the increased presence of Tregs using a novel method. We measured the fractal dimensions (FD) and lacunarity (LC) of Tregs in MT and WT KRAS NSCLC tumor specimens to discern their enrichment. Fractals are infinitely complex patterns that are self-similar across different scales. Objects that exhibit exact, quasi, or statistical self-similarity may be considered fractal. LC is indicative of a fractal with large gaps, and the converse is true for low LC. Thus, immunohistochemistry (IHC) staining with three markers (DAPI, CD4, and FOXP3) was performed. The DAPI-stained nuclei were used to identify the overall cells on the tissue. CD4- and FOXP3-stained images were used to identify T-helper cell and Treg cell distribution in the tumor microenvironment. The IHC images were segmented to discern cells in the tissue sections, as described in the Experimental Section. As anticipated, this analysis showed that there was an enrichment of the FOXP3^+^ population in samples derived from MT KRAS tumors when compared to WT KRAS tumors (Appendix A).

## 4. Discussion

A major hurdle for effective cancer immunotherapy is the immunosuppressive tumor microenvironment. Although large numbers of tumor-specific T cells can be generated in patients by active immunization or adoptive transfer, these T cells do not readily translate to tumor cell killing in situ. In fact, the Treg subpopulation of T cells plays an important role in suppressing tumor-specific immunity [31,32]. Thus, our observations (Appendix A) employing FD/LC analyses of clinical samples from patients with MT KRAS NSCLC had a higher percentage of FOXP3^+^ Treg-like cells underscore the significance of the present study.

We used a variety of experimental approaches to demonstrate that mutant KRAS TDEs from lung cancer cell lines, patient serum, and xenograft mice can actuate phenotypic switching in T cells independent of cytokines produced by the tumor cells. Furthermore, we demonstrated that in addition to the molecular markers that characterize Tregs, converted cells have a metabolic profile that parallels bona fide Tregs and also function like authentic Tregs by suppressing proliferation of naïve T cells. However, despite these similarities, single-cell sequencing data revealed that the conversion results in a new class of immune regulatory cells that share many similarities with bona fide Tregs. Our data are consistent with previous observations demonstrating phenotypic and functional diversity in human Treg cells [33]. In fact, such heterogeneity may allow subsets of Tregs with unique specificities and immunomodulatory functions to be targeted to define immune environments during different types of inflammatory responses [33]. Additional studies aimed at discerning the function(s) of this new class of Tregs are needed to fully appreciate their role in tumorigenesis. 

Nonetheless, the present study raises some important questions. How does mutant KRAS cause phenotypic switching in T cells? One potential mechanism by which mutant KRAS could modulate FOXP3 expression is by acting at the transcriptional level. Consistent with this possibility, it has been reported that H-RAS is localized to the nucleus [34,35]. However, it should be noted that another study that detected KRAS in the nucleus pointed to potential limitations imposed by the non-specificity of the antibodies or contaminations in cellular preparations [36]. However, our attempts employing immunofluorescence microscopy of green fluorescence protein (GFP) WT KRAS fusion protein or MT KRAS GFP fusion protein did not show nuclear localization of either protein.

An alternate possibility is that KRAS could regulate FOXP3 expression without entering the nucleus via aberrant signaling. For example, oncogenic N-RAS was shown to act as the most potent regulator of SRF-, NF-*κ*B-, and AP-1-dependent transcription. The N-RAS and RGL2 (Ral guanine nucleotide exchange factor) axis is a distinct signaling pathway for SRF-targeted gene expression such as Egr1 and JunB as the RGL2 RAS binding domain significantly impairs oncogenic N-RAS-induced SRE activation. Indeed, oncogenic N-RAS elevated acetylated histone H3K9 and H3K23 levels globally in the chromatin, and chromatin immunoprecipitation assays revealed that acetylated H3K9 is significantly enriched at the promoter and coding regions of Egr1 and JunB [37].

Our single-cell sequencing results suggest that MT KRAS may also modulate phenotypic switching without actually entering the nucleus (Appendix A). To this end, we identified an IFN gene expression signature induced by MT KRAS in CD4^+^ naïve T cells. Since WT KRAS did not induce the expression of the same genes, we conclude that MT KRAS is responsible for upregulation of type I IFN expression. This conclusion is further corroborated by the fact that the transcription factor IRF7, a key transcriptional regulator of type I IFN-dependent immune responses, was upregulated in the converted cells but not in the naïve T cell control Group. Other groups have also found a link between different types of IFNs (including Type I) and FOXP3 expression in mice and humans [28,29,30]. Although more work will be needed to fully understand the exact mechanism linking mutation of KRAS and cell signaling, leading to IFN expression and phenotypic conversion of CD4^+^ naïve T cells to Tregs, we propose a working model (Appendix A). TDEs shed from NSCLC cells carrying MT KRAS are incorporated into CD4^+^ naïve T cells and induce the IRF7 signaling pathway. This leads to increased gene expression and induction of key proteins involved in expression and secretion of type I IFN. Type I IFN, in turn, induces the conversion of CD4^+^ naïve T to Treg-like cells by upregulating genes that characterize Tregs, including FOXP3. However, we note that the switch is not a binary event but a gradual process that appears to involve a series of bifurcations. Consequently, MT KRAS-mediated conversion results in a heterogeneous population with varying Treg-like phenotypes with cells that have a close resemblance to bona fide Tregs, expressing the full complement of Treg-restricted genes including FOXP3.

Another possibility is that the KRAS protein could upregulate FOXP3 protein post-transcriptionally, as has been reported with the zinc finger protein ZNF304 [38]. A majority of KRAS-positive colorectal cancers (CRCs) have a CpG island methylator phenotype (CIMP). CIMP is characterized by aberrant DNA hypermethylation and transcriptional silencing of many genes that are encoded by the INK4-ARF locus. An RNA interference screen identified the zinc-finger DNA-binding protein ZNF304 as the pivotal factor required for INK4-ARF silencing and CIMP. In these cells, ZNF304 is bound at the promoters of INK4-ARF and other CIMP genes and recruits a corepressor complex that includes the DNA methyltransferase DNMT1, resulting in DNA hypermethylation and transcriptional silencing. Thus, KRAS promotes silencing through upregulation of ZNF304. Therefore, it is conceivable that KRAS may promote upregulation of the transcription factor(s) (e.g., NFAT) [39] that, in turn, binds to the FOXP3 promoter and drives its expression in T cells and thereby, activates the cascade of events, culminating in their conversion to Treg-like cells. Additional studies that are currently underway in our laboratory could help uncover the identity of this new class of converted T cells in NSCLC.

The present study also begs the question as to why only MT KRAS, but not WT KRAS, can actuate phenotypic switching in T cells? Indeed, this is related to the more general question of why only mutant KRAS is oncogenic, and, to date, this remains one of the most challenging questions in cancer biology. Perhaps some recent biophysical and computational studies may help shed new light. A study investigating the effect of point mutations on the structure of KRAS employing UV photo disassociation mass spectrometry [40] revealed that different downstream effects occur due to differences in the long-range conformational dynamics specific to each of the commonly occurring KRAS point mutations. Similarly, a molecular dynamics study based on conditional time-delayed correlations, identified driver–follower relationships in correlated motions of KRAS residue pairs, revealing the direction of information flow during allosteric modulation of its nucleotide-dependent intrinsic activity. Thus, the more C-terminal active Switch-II region motions drive the more N-terminal Switch-I region motions in the KRAS molecule [41]. Interestingly, the short region of the C-terminal region of KRAS (residues 167–188/189, also called the hypervariable region) is intrinsically disordered [42]. Thus, in MT KRAS, intrinsic disorder appears to facilitate intramolecular interactions spanning long distances that appear to be crucial for activation mechanisms and intensified oncogenic signaling compared to WT KRAS [40].

Lastly, it is unknown whether the phenotypic switch is transient or permanent, i.e., can be inherited. It is important to note that biological barriers may likely curtail a permanent horizontal transformation of normal cells through such a TDE-mediated mechanism [12]. Although this issue is outside the scope of the present study, it is possible that the mutant KRAS-actuated phenotypic switch we observe with lung cancer-derived exosomes is transient and the number of T cells converted would depend on the tumor mass. Consistent with this argument, our in vivo studies showed that TDEs from control mice that harbored large tumors were able to convert a larger population of T cells compared to TDEs from *Salmonella*-infected mice with attenuated tumors. As MT KRAS is present in 41% of NSCLC patients [43], this newly discovered MT KRAS-induced phenotypic switching could have important therapeutic implications for lung cancer. However, given that KRAS is the most frequently mutated oncogene in a wide variety of cancers, the present work is likely to have a much wider impact. 

## 5. Conclusions

In this paper we have uncovered a hitherto forth unappreciated function for oncogenic KRAS. We have shown that MT KRAS promotes phenotypic conversion of CD4^+^ T cells to FOXP3^+^ cells, and like regular Tregs, the converted cells possess an immune-suppressive function. However, at a molecular level, the KRAS-converted FOXP3^+^ cells appear to be distinct from regular Tregs. Furthermore, we identified the interferon pathway as the mechanism underlying the phenotypic switch. These observations highlight a novel cytokine-independent, cell-extrinsic role for KRAS in T cell phenotypic switching. Thus, targeting this new class of Treg-like cells discovered in the present work represents a unique therapeutic approach for KRAS-driven cancers including non-small cell lung cancer. 

## Figures and Tables

**Figure 1 jcm-08-01726-f001:**
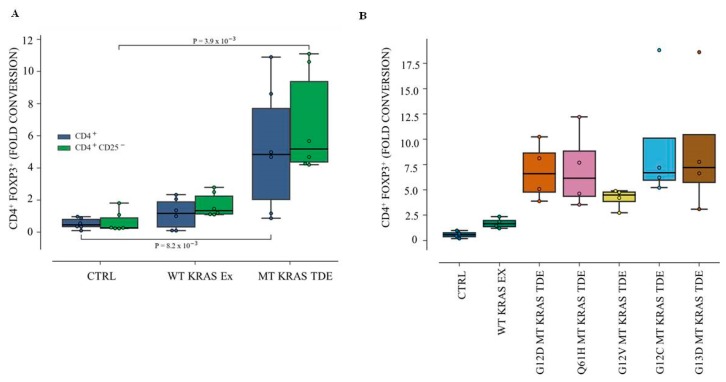
Phenotype conversion of CD4^+^ and naïve CD4^+^ CD25^−^ T cells to CD4^+^ FOXP3^+^ Treg-like cells by tumor-derived exosomes from wild type and mutant KRAS lung cancer cells and patient serum. CD4^+^ and naïve CD4^+^ CD25^−^ T cells were isolated from healthy donor PBMCs and incubated with wild type (WT KRAS Exo) or Q61H mutant KRAS (MT KRAS Exo) tumor TDEs for 24 h. (**A**) Blue box plots represent fold conversion of CD4^+^ T cells incubated with MT KRAS or WT KRAS TDEs to CD4^+^ FOXP3^+^ cells, as analyzed by flow cytometry. Green box plots represent fold conversion of naïve CD4^+^ CD25^−^ T cells incubated with MT KRAS or WT KRAS TDEs to CD4^+^ FOXP3^+^ cells. Numbers represent *p*-values. (**B**) MT KRAS TDEs (G12D, Q61H, G12V, G12C, and G13D) were isolated from patient serum and incubated with naïve CD4^+^ CD25^−^ T cells isolated from donor PBMCs, and fold conversion to CD4^+^ FOXP3^+^ cells was determined by flow cytometry. Data are representative of 4–5 independent experiments. Box plots represent the inter-quartile range (IQR), where the horizontal line indicates the median. Whiskers extend to the farthest data point within a maximum of 1.5 × IQR. All *p*-values were calculated using the Mann–Whitney *U*-Test.

**Figure 2 jcm-08-01726-f002:**
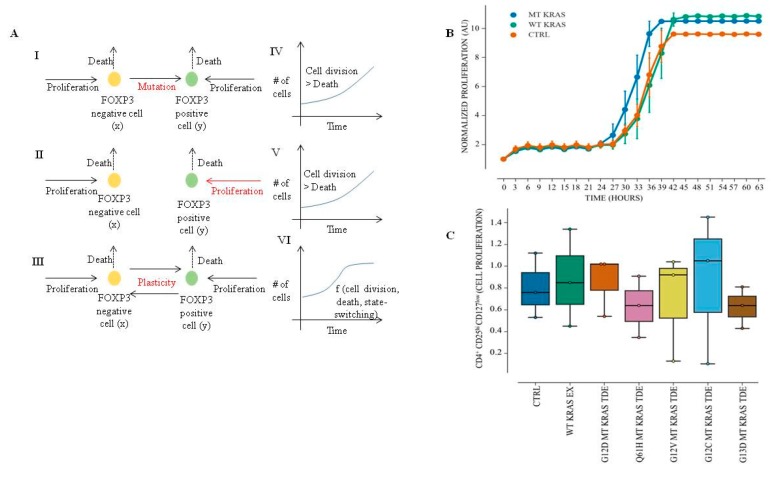
Mathematical modeling of phenotypic switching of T cells to Treg-like cells. (**A**) I–III schematics representing various scenarios that have been mathematically modeled. IV–VI representative outcomes of three scenarios (respectively presented in the same row) showing that the cell population continues to grow in IV–V, but tends to reach a steady state in VI. (**B**) Naïve CD4^+^ CD25^−^ T cells were incubated with MT or WT KRAS TDEs for 24 h. Cell proliferation was analyzed using the IncuCyte Live Cell Imaging System. Graph shows the mean and standard deviation of duplicate wells. (**C**) CD4^+^ CD127^low^ CD25^+^ cells isolated from healthy donor PBMCs were incubated for 24 h with WT or Q61H MT KRAS TDEs isolated from the serum of NSCLC cancer patients, and cell proliferation was assessed using the IncuCyte Live Cell Imaging System. Box plots represent the interquartile range (IQR), with the horizontal line indicating the median. Whiskers extend to the farthest data point within a maximum of 1.5 × IQR. All p-values were calculated using the Mann–Whitney *U*-test.

**Figure 3 jcm-08-01726-f003:**
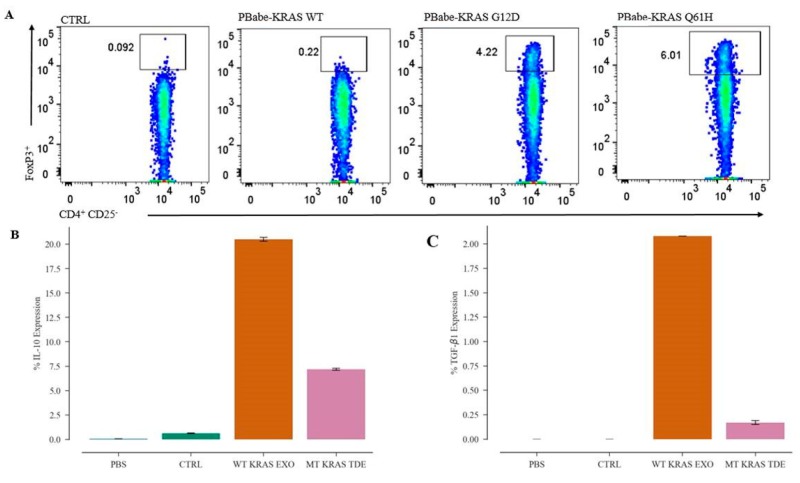
MT KRAS cDNA, in the absence of other biomolecules, is sufficient to convert T cells to Tregs. (**A**) Naïve CD4^+^ CD25^−^ T cells were transfected with plasmid PBabe-KRAS WT, PBabe-KRAS G12D, or PBabe-KRAS Q61H MT, and the percentage of CD4^+^ FOXP3^+^ cells was determined by flow cytometry. Data from the four independent donor samples are shown as a representative flow cytometry dot plot obtained by gating total viable CD4^+^ or CD4^+^ naïve T cells transfected with the WT or MT KRAS plasmid. (**B**,**C**) Attune NXT flow cytometry was used to evaluate percent expression of the intracellular cytokines (**B**) IL-10 and (**C**) TGF-β1 derived from WT and MT KRAS TDEs.

**Figure 4 jcm-08-01726-f004:**
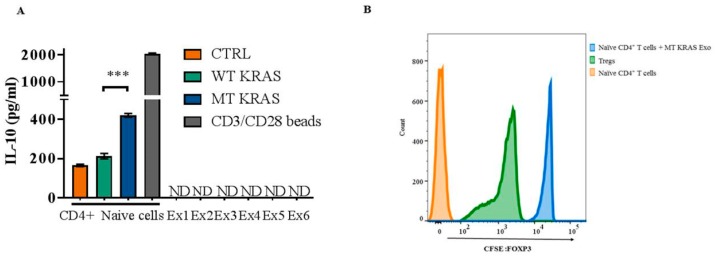
MT KRAS-mediated phenotypic switching of T cell to Treg-like cells is independent of IL-10, and the converted Treg-like cells can suppress naïve CD4 T cell proliferation. (**A**) MT KRAS (Ex1–4) and WT KRAS (Ex5–6) TDEs isolated from the serum of NSCLC patients or naïve CD4^+^ CD25^−^ T cells incubated with WT or MT KRAS TDEs for 24 h were evaluated for release of IL-10 via enzyme-linked immunosorbent assays (ELISA). CD3/CD28 beads were used as a positive control. Graph shows the mean and standard deviation of quadruple wells. ND, not detectable. *** *p* < 0.001 (two-tailed unpaired t-test). (**B**) Carboxyfluorescein succinimidyl ester (CFSE)-labeled naïve CD4^+^ CD25- T cells isolated from healthy donor PBMCs were incubated with MT KRAS or WT KRAS TDEs for 24 h, and cell proliferation was assessed by flow cytometry. The plot shows the cell count calculated as the fluorescence intensity based on gating cells positive for CFSE dye. As a positive control, CFSE-labeled naïve CD4^+^ CD25^−^ T cells were incubated with CD4^+^ CD25^hi^ CD127^low^ Tregs isolated from the same donor.

**Figure 5 jcm-08-01726-f005:**
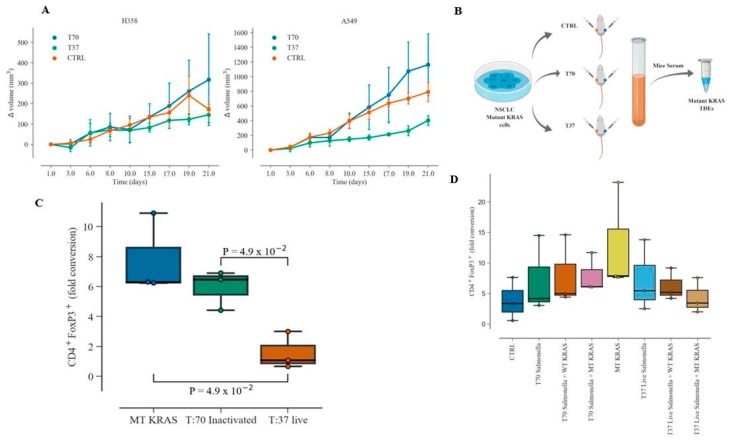
TDEs from mice harboring exponentially growing tumors convert T cells to Treg-like cells. Mouse xenografts with MT KRAS NSCLC cells harboring the G12D (A549 cells) or the G12C (H358 cells) KRAS mutations were created in NSG mice (*n* = 12) to initiate a two-sided xenograft model. (**A**) Mice were randomly assigned to three groups: (1) control (CTRL) group, (2) infected with heat-inactivated Salmonella (T70) group, and (3) infected with live bioluminescent Salmonella typhimurium YS1646 (T37) group. (**B**) At day 21, mice were euthanized and exosomes were isolated. (**C**) Naïve CD4^+^ CD25^−^ T cells from donor PBMCs were treated with MT KRAS TDEs isolated from mice sera, with induced tumor, infected with heat-inactivated Salmonella (T70), or live bioluminescent Salmonella typhimurium. Fold conversion of naïve CD4^+^ CD25^−^ T cells to CD4^+^ FOXP3^+^ Treg-like cells was assessed by flow cytometry. (**D**) The in vitro effect of live (37 ˚C) or heat-inactivated (T70) bioluminescent Salmonella typhimurium on naïve CD4^+^ CD25^−^ T cells incubated with WT or MT KRAS TDEs was evaluated. The box plot for the three independent donor samples shows fold conversion to CD4^+^ and FOXP3^+^ and represents the interquartile range (IQR) where the horizontal line indicates the median. Whiskers extend to the farthest data point within a maximum of 1.5 × IQR. All *p*-values were calculated using the Mann–Whitney *U*-Test.

**Figure 6 jcm-08-01726-f006:**
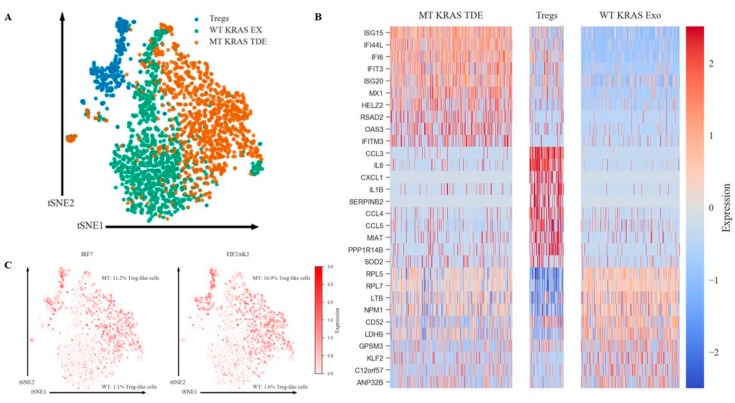
MT KRAS-mediated phenotypic switching results in conversion of T cells to Treg-like cells that are distinct from bona fide Tregs. (**A**) t-SNE clustering of CD4^+^ naïve T cells treated 24 h with WT or MT KRAS exosomes (ex) and Tregs from the same donor. (**B**) Heat maps representing the top ten signature genes from CD4^+^ naïve T cells treated with WT or MT KRAS TDEs and Tregs. (**C**) t-SNE projection of gene expression patterns in all groups. Percent (%) representing the proportion of MT KRAS cells having similar genes expression levels to Tregs cells.

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
