# Peer review of "Phenotypic Switching of Naïve T Cells to Immune-Suppressive Treg-Like Cells by Mutant KRAS"

_jcm, 2019, doi:10.3390/jcm8101726_

Round 1
Reviewer 1 Report
In the current the authors demonstrate the tumour derived exosomes of mutant (MT) KRAS non-small cell lung cancer can drive phenotype switching to FOXP3 T-reg phenotype from naive T cells. The authors also present data that suggests this is through through manipulation of the type I interferon pathway rather than cytokine stimulation.
Overall the manuscript is well written and organised in a very accessible way. The data is well presented and supports the conclusion made. I have no additional comments.
Author Response
Reviewer #1 had no specific comments.
Reviewer 2 Report
The manuscript “Phenotypic Switching of Naïve T cells to Immune-suppressive Treg-like Cells by Mutant KRAS” by Kalvala A et al. focused on a relevant question about the immunomodulation of T cells by TDEs secreted by NSCLC bearing WT or mutant K-RAS.
The authors showed interesting results but there are some comments:
-In the Discussion section the authors reported that it is unknown whether the phenotypic switch of Naïve T cells is transient or permanent. To give force to the theory of the mathematical model III (plasticity between Tregs and non-Tregs) rather than increased proliferation of pre-existing Tregs, they should demonstrate a reversible phenotypic switch between non-Tregs to Tregs thus showing a loss of Tregs phenotype overtime.
- The authors could perform a time course incubation assay with TDEs bearing mutant and WT K-Ras in order to assess a time dependent modulation of naïve T cells by TDEs. Furthermore they also should isolate TDEs after the incubation time and test the maintenance of the Treg-like phenotype overtime.
-In the Results section at paragraph 3.4. the authors describes a flow cytometry analysis of Ki 67 levels in CD4+ T cells incubated with WT and MT KRAS TDEs showing no significant changes in Ki67 expression between all stained populations. They concluded that the increase in CD4+ FOXP3+ Treg-like cell population was indeed due to phenotypic conversion rather than an increase in proliferation of pre-existing Treg-like cells. I think that the authors should describe these results as an unchanged proliferation rate in the Treg-like cells in comparison to control or CD4+ T cells incubated with WT KRAS TDEs rather than an evidence of no increased proliferation of pre-existing Treg-like cells.
-The author also describe an enrichment of the tumor microenvironment in Tregs. Did the authors analyse any stroma cells to test the TDEs internalizations and the presence of the delivered DNA also in such type of stroma population?
Author Response
Reviewer # 2
In the Discussion section the authors reported that it is unknown whether the phenotypic switch of Naïve T cells is transient or permanent. To give force to the theory of the mathematical model III (plasticity between Tregs and non-Tregs) rather than increased proliferation of pre-existing Tregs, they should demonstrate a reversible phenotypic switch between non-Tregs to Tregs thus showing a loss of Tregs phenotype overtime.
Author’s response: This a good point and we appreciate it. Indeed, we attempted to demonstrate the reversible switch by culturing the converted Tregs and monitoring FOXP3+ expression over time. While we observed the decline, we were unable to culture the converted cells in vitro for several days to discern a significant drop and therefore, were unable to report these preliminary results.
The authors could perform a time course incubation assay with TDEs bearing mutant and WT K-Ras in order to assess a time dependent modulation of naïve T cells by
TDEs. Furthermore they also should isolate TDEs after the incubation time and test the maintenance of the Treg-like phenotype overtime.
Author’s response: We appreciate the reviewer’s comment. In preliminary experiments we determined FOXP3+ expression at earlier time points (6 and 12 h) but did not observe expression determined by flow sorting. Conversely, at 48 h, there was no further change in FOXP3+ levels. However, we are unclear what the reviewer means by, “Furthermore they also should isolate TDEs after the incubation time and test the maintenance of the Treg-like phenotype overtime.” We are not sure how isolating TDEs from converted cells will inform one about the status of phenotype over time?
In the Results section at paragraph 3.4. the authors describes a flow cytometry analysis of Ki 67 levels in CD4+ T cells incubated with WT and MT KRAS TDEs showing no significant changes in Ki67 expression between all stained populations.
They concluded that the increase in CD4+ FOXP3+ Treg-like cell population was indeed due to phenotypic conversion rather than an increase in proliferation of preexisting Treg-like cells. I think that the authors should describe these results as an unchanged proliferation rate in the Treg-like cells in comparison to control or CD4+ T cells incubated with WT KRAS TDEs rather than an evidence of no increased proliferation of pre-existing Treg-like cells.
Author’s response: We agree with reviewer and have accordingly added text to reflect this alternate interpretation in the revised manuscript.
The author also describe an enrichment of the tumor microenvironment in Tregs. Did the authors analyse any stroma cells to test the TDEs internalizations and the presence of the delivered DNA also in such type of stroma population?
Author’s response: This is an interesting idea and we thank the anonymous reviewer for the pointing this out. However, we feel this may out of scope of the present endeavour and will be something we will take up as part of our ongoing research in this area.
Reviewer 3 Report
Comments to the Authors
General Comments: In this manuscript by Kalvala and colleagues, the authors provide a very interesting dataset demonstrating that mutant KRAS tumor-derived exosomes (TDE) can induce expression of FOXP3 in CD4+CD25- lymphocytes. The authors employ several experimental methods to evaluate this phenotypic change. Through these experiments they find that mutant KRAS elicits increased phenotypic conversion to a FOXP3 positive state. Then, with a mouse xenograft tumor model they demonstrate that TDEs obtained from attenuated tumor growths have altered rates of conversion compared to TDEs from tumors with unaltered growth rates. Additionally, they performed single-cell sequencing and identified interferon signaling as a potential mechanism for the mutant KRAS TDE phenotypic conversion to a Treg like state. The authors suggest that upregulation of type I interferons is one potential mechanism for the phenotypic conversion. There are several comments.
Comments:
In the methods section the authors do not list the clone for the FOXP3 antibody. In humans there are several FOXP3 splice variants, unlike in mice, and while the antibody clone used likely detects both the full length and the delta exon 2 variant knowing which clone and its epitope should be provided. Furthermore, there is data that the full length variant is expressed under activating conditions and determined the ratio between full length and delta exon 2 may support their hypothesis. It is also unclear to this reviewer in the methods if the amount of exosomes is standardized between the experiments. For example, by number of particles or total protein concentration. In the methods the authors do not list the infectious dose or cite a reference for the Salmonella infection experiments and also for how the Salmonella supernatants were collected and standardized. For Figure 1A, are the WT KRAS exosomes from the H1993 human NSCLCs? For Figure 1B, p-values and statistics are not provided in the figure. In regards to the cell proliferation and suppression assays, did the authors attempt cell activation with anti-CD3/CD28 beyond the control for figure 4. To better support the findings that converted CD4+ T cells can suppress proliferation a more standard Treg suppression assay should be employed as without stimulation CD4+ cells will have low proliferation rates. Furthermore, the CFSE-labeled naïve CD4+ T cells unstimulated would be expected to retain some CSFE; however in Figure 4B these appear CSFE negative. For Figure 4B is the x-axis gated for CSFE and FOXP3 expression or just CSFE? On page 10, line 408 and 412 do the authors mean to write: CD4+ CD25hiCD127low (FOXP3+) instead of “CD4+ FOXP3+ CD127low (FOXP3+).” On page 12 for the metabolic profiling the authors provide data in the supplement that utilization of carbon sources were significantly reduced in the MT KRAS TDEs converted T cells and Tregs. While the numbers in Table S2 are lower compared to Naïve CD4 T cells there are no statistics to support a significant reduction. For Figure 5D, p-values are not provided, are these changes significant? Similarly for Figure S7, the fractal analysis and lacunarity data is very interesting however was the enrichment stated in the results actually significant as there are no statistics provided in the figure or figure legend. On page 15, line 584 do the authors mean Figure S7 instead of Figure S8? If a potential mechanism is translocation of IRF7 to the nucleus in MT KRAS TDEs, have the authors looked for this potential difference between conditions?Author Response
Reviewer # 3
In the methods section the authors do not list the clone for the FOXP3 antibody. In humans there are several FOXP3 splice variants, unlike in mice, and while the antibody clone used likely detects both the full length and the delta exon 2 variant knowing which clone and its epitope should be provided. Furthermore, there is data that the full length variant is expressed under activating conditions and determined the ratio between full length and delta exon 2 may support their hypothesis.
Author’s response: The antibody used, anti‐human mAb FOXP3‐PE antibody was against the full-length FOXP3 protein and not the variant, clone ID# (206D) catalog#320107 from Biolegend.
It is also unclear to this reviewer in the methods if the amount of exosomes is standardized between the experiments. For example, by number of particles or total protein concentration.
Author’s response: We used equal number of exosomes standardized between each experiment that we measured by NanoSight as mentioned in Methods section 2.4. We used 1x106/ml of particles per preparation for both wild type and mutant KRAS cells. This is mentioned in the Supplementary Fig. S1 figure legend for NanoSight analysis and highlighted in yellow.
In the methods the authors do not list the infectious dose or cite a reference for the Salmonella infection experiments and also for how the Salmonella supernatants were collected and standardized.
Author’s response: To address the reviewer’s concerns, we amended the original text and highlighted the new text in yellow.
For Figure 1A, are the WT KRAS exosomes from the H1993 human NSCLCs? For Figure 1B, p-values and statistics are not provided in the figure.
Author’s response: That is correct for Figure 1A. The p values in Figure 1B were omitted because they were not statistically significant.
In regards to the cell proliferation and suppression assays, did the authors attempt cell activation with anti-CD3/CD28 beyond the control for figure 4. To better support the findings that converted CD4+ T cells can suppress proliferation a more standard Treg suppression assay should be employed as without stimulation CD4+ cells will have low proliferation rates. Furthermore, the CFSE-labeled naïve CD4+ T cells unstimulated would be expected to retain some CSFE; however in Figure 4B these appear CSFE negative. For Figure 4B is the x-axis gated for CSFE and FOXP3 expression or justCSFE? On page 10, line 408 and 412 do the authors mean to write: CD4+ CD25 CD127 (FOXP3+) instead of “CD4+ FOXP3+ CD127 (FOXP3+).”
Author’s response:
We used CD3/CD28 activation beads for ELISA based method to detect IL-10 secretion (Fig. 4A), since exosomes have low levels of IL-10 as compared with CD3/CD28 beads. Following incubation with exosomes, they were analyzed by flow cytometry. Since incubation with CD3/CD28 causes activation giving rise to a greater number of FOXP3+ Tregs, they cannot be used as control for comparison with FOXP3-like Tregs generated upon incubation with exosomes. Further the observed FOXP3-like Treg phenotype is mutant KRAS derived which is different from naïve or induced Tregs which one can measure upon CD3/CD28 activation. The main aim was to identify FOXP3-like Tregs that suppress proliferation of naïve CD4 CD25 negative cells upon incubation with MT KRAS exosomes rather than activation using CD3/CD28 that usually increases proliferation without the stimulation. We used CFSE negative cells along with CD4 CD25 hi CD127 low FOXP3 Tregs as positive to analyze the suppression of CD4 naïve T cell proliferation. We have mentioned the suppression assay in Materials and Methods section.
On page 10, line 408 and 412 do the authors mean to write: CD4+ CD25 CD127 (FOXP3+) instead of “CD4+ FOXP3+ CD127 (FOXP3+)?Author’s response: Yes, and this has been corrected (and highlighted in yellow).
On page12 for the metabolic profiling the authors provide data in the supplement that utilization of carbon sources were significantly reduced in the MT KRAS TDEs converted T cells and Tregs. While the numbers in Table S2 are lower compared to Naïve CD4 T cells there are no statistics to support a significant reduction.
Author’s response: The microarray phenotype assay was performed using the Biolog pre-array plate following the manufacturer’s protocol. All the numbers reported in the Table S2 are based on the percent intensity of the redox dye coloration due to generation of NADH. For reference we cited, Angelin A, Gil‐de‐Gómez L, Dahiya S, Jiao J, Guo L, Levine MH, Wang Z, Quinn WJ 3rd, Kopinski PK,
Wang L, Akimova T, Liu Y, Bhatti TR, Han R, Laskin BL, Baur JA, Blair IA, Wallace DC, Hancock WW, Beier UH. Foxp3 Reprograms T Cell Metabolism to Function in Low‐Glucose, High‐Lactate Environments. Cell Metab. 2017 25(6):1282‐1293.e7 [ref #20].
For Figure 5D, p-values are not provided, are these changes significant?
Author’s response: The p values were not statistically significant.
Similarly for Figure S7, the fractal analysis and lacunarity data is very interesting however was the enrichment stated in the results actually significant as there are no statistics provided in the figure or figure legend.
Author’s response: The difference of the fractal dimension and lacunarity between KRAS mutant samples and wild type are statistically significant. We added the p values to the boxplots in the revised manuscript. The p values were calculated using the Mann-Whitney U-test. The detailed analysis is described in the manuscript section 2.24.
On page 15, line 584 do the authors mean Figure S7 instead of Figure S8? If a potential mechanism is translocation of IRF7 to the nucleus in MT KRAS TDEs, have the authors looked for this potential difference between conditions?
Author’s response: We don’t believe that is true, and unfortunately, we did not look into IRF7 translocation mechanism as yet. They are part of our future plans.
Round 2
Reviewer 2 Report
-The authors well address each points of previous review, but they are not able to demonstrate plasticity between Tregs and non-Tregs, hence, in order to strongly support the conclusion that the expansion of the Treg-like population upon exposure to MT KRAS TDEs is due to phenotypic switching rather than the increased proliferation of pre-existing Tregs or the exogenous DNA incorporation into the genoma of the recipient cell, the authors should perform additional experiments.
-Furthermore, in the point two of previous review, with the sentence “Furthermore they also should isolate TDEs after the incubation time and test the maintenance of the Treg-like phenotype overtime”, I mean to check the maintenance of the Treg-like phenotype in T cells after removing of TDEs from the cell culture. However, this question has been addressed by the authors in the author’s response to point 1 of previous review.
Author Response
The authors well address each points of previous review, but they are not able to demonstrate plasticity between Tregs and non-Tregs, hence, in order to strongly support the conclusion that the expansion of the Treg-like population upon exposure to MT KRAS TDEs is due to phenotypic switching rather than the increased proliferation of pre-existing Tregs or the exogenous DNA incorporation into the genoma of the recipient cell, the authors should perform additional experiments.
Author’s response: We appreciated the comments/suggestions that the Reviewer made in the first revision. They were constructive criticisms and we provided in-depth responses. The MS explains how we arrived at the three mathematical models to rule out the possibility that the phenomenon was indeed due to phenotypic conversion and not due to the proliferation of existing Tregs. Furthermore, we also explained in the response to the reviewer the difficulty associated with growing isolated T cells in culture beyond 48 h (which is known from other previous studies) Given that the conversion of T cells to Treg-like cells by MT KRAS is the major finding that we would like to report, we went to enormous lengths to demonstrate this observation in the MS. Based on this, we believe the reviewer’s request that we demonstrate that the expansion of the Treg-like population upon exposure to MT KRAS TDEs is due to phenotypic switching rather than the increased proliferation of pre-existing Tregs is experimentally fraught with errors in the current systems that are available. We have incorporated this in the manuscript and appreciate it very much.
Furthermore, in the point two of previous review, with the sentence “Furthermore they also should isolate TDEs after the incubation time and test the maintenance of the Treg-like phenotype overtime”, I mean to check the maintenance of the Treg-like phenotype in T cells after removing of TDEs from the cell culture. This question has been addressed by the authors in the author’s response to point 1 of previous review.Author’s response:
Per the Reviewer, this issue has already been addressed in our previous response.
Reviewer 3 Report
The authors have addressed the comments.
Author Response
This reviewer was satisfied with our response. No additional comments were made by this reviewer.